



# A Q methodological approach to identify practitioners' viewpoints on citizen science in Dutch regional water resource management

E. Minkman[1,2], M.M. Rutten[1], M.C.A. van der Sanden[2]

[1]Department of Water Resource Management, Delft University of Technology, Delft, 2628 CN, The Netherlands
[2]Department of Science Education & Communication, Delft University of Technology, Delft, 2628 CJ, The Netherlands

*Correspondence to*: E. Minkman (minkman.ellen@gmail.com)

**Abstract.** Citizen science as a data collection method has gained popularity in various scientific fields and is considered by many as a potential means of effective science communication. In recent years, both practitioners and governments have started to use citizen science as a form of public participation. The governing body of the Dutch Water Authorities considers citizen science a possible solution for helping bridge the water awareness gap among the general public in order to better manage the pressures on the water governance system. The motivation of experts, and in particular practitioners, to engage in citizen science has seldom been studied. This article aims to pinpoint the various viewpoints of practitioners on citizen science in water quality monitoring at Dutch regional water authorities. A Q methodological approach was used because it allows a statistical analysis to be conducted with a small sample size. 33 practitioners at eight different water authorities ranked 46 statements from agree to disagree. Three factors were extracted using a factor analysis and were transformed into three narrative viewpoints. These viewpoints are: 1) understanding citizen science as a potential solution for achieving various goals, thereby encouraging citizen participation in data collection and analysis; 2) considering citizen science a method for additional, illustrative data; and 3) viewing citizen science primarily as a means of education. These viewpoints show practitioner's support for citizen science, although no support for higher levels of citizen engagement were found. These findings not only demonstrate the potential interest of the Dutch water authorities in using citizen science, but the identified viewpoints can be used to enhance the design of citizen science projects.

## 1 Introduction

Citizen science has gained popularity over the past twenty years, which started in the fields of ecology and astronomy (Cohn 2008). Citizen science involves citizens' actively in collecting, analysing or interpreting data. The British Trust of Ornithology (e.g. Hobbs & White 2012) has been collecting bird population data by citizen scientists over large geographical areas for the past century. This count is similar to the American Christmas Bird Count (e.g. Cohn 2008) and the Dutch Tuinvogeltelling (garden bird count). The Galaxy Zoo project is an online citizen science campaign where citizens classify pictures of galaxies (e.g. Raddick et al. 2010). Other scientific fields have also adopted citizen science over the past decades



(Cohn 2008). Most citizen science initiatives focus on goals of education and knowledge generation, which makes effective science communication possible (Varner 2014).

The use of citizen science has become common practice in many scientific fields so that conservation professionals
(including those working at conservation organizations, such as nature managers), and government practitioners have started to realize the potential use of citizen science as well. Government practitioners are defined as those working at a government agency or at the local government level. Conservation professionals have also adopted the principles of citizen science, but they tend to emphasize raising awareness, knowledge generation and capacity building (Weng 2015). The information collected through citizen science is most often used for science-based or evidence-based management. Although government
practitioners have shown an increasing interest in citizen science as well, available examples have been  infrequently documented in peer-reviewed journals (Conrad & Hilchey 2011).

The interest in citizen science is also growing among those involved in Dutch water management. The Dutch government, the umbrella association of Dutch regional water managers, and the Dutch Water Authorities (Unie van Waterschappen, UvW) have mentioned public participation as a potential solution to current and future challenges (Tielrooij 2000; UVW
21015). According to the Organisation of Economic Cooperation and Development (OECD), Dutch water management is an international example, but Dutch citizens reveal a "striking awareness gap" on water related issues (OECD 2014). The Netherlands currently faces issues related to climate change, urbanisation and an increasing financial pressure in funding its regional water infrastructure.

Citizen science is considered effective for a certain target audience and for certain levels of interaction (Varner 2014). Citizen involvement is often limited to data collection or so-called contributory projects (Bonney et al. 2009). There are some examples in which citizens are involved in the analysis of samples or analysing results (collaborative projects), as well as projects where citizens participate in other aspects of the research process, such as defining research questions or
translating results into action (co-created projects) (Bonney et al. 2009).

Citizens' motivations have been studied extensively in a diverse set of citizen science projects, such as online crowdsourcing (e.g. Raddick et al. 2010; Rogstadius et al. 2011; Chandler & Kapelner 2013) and environmental monitoring (e.g. Hobbs & White 2012; Roy et al. 2012; Edwards 2014). It has been acknowledged that the idea of 'the public' does not really exist
(e.g. Varner 2014), since 'the public' consists of a wide variety of people with different backgrounds, interests, traits, values and beliefs. Nevertheless, existing studies of (online) citizen science, despite this diversity, reveal the same dominant motivations over a wide range of projects and participants. Citizens engage in citizen science because they think it is fun, because the topic interests them and because they want to contribute to science or nature conservation (e.g. Raddick et al. 2010; Rogstadius et al. 2011; Hobbs & White 2012; Roy et al. 2012; Chandler & Kapelner 2013; Edwards 2014). Citizens





are motivated to continue to contribute by: (increasing) the extent of their involvement (Rotman et al. 2012; Roy et al. 2012), offering feedback concerning the work at three levels (individual contribution, group contribution and the use of data) and building a relationship based on trust between scientists and citizens (Rotman et al. 2012).

However, the motivational drivers behind the experts' use of citizen science (scientists, conservation professionals and government practitioners, in particular) have been less frequently studied and the studies available report different findings. Scientist's motivations are primarily to advance science as well as develop their careers (Rotman et al. 2012). This is compatible to a citizen's motivational desire to contribute to science and conservation or to engage in exploring a topic of their interest further (e.g. Rotman et al. 2012; Edwards 2014). In contrast, Weng (2015) identifies three areas of friction

between the vision of scientists and the volunteers with regard to citizen science. The first is the short-term participation of volunteers that conflicts with scientists' interest in long-term processes. The second concerns the limits of what volunteers can do and their dissatisfaction with the research processes. The third regards a power hierarchy between citizens and scientists. Rotman et al. (2012) found that while the motivations of citizens and scientists are complementary, they can also change over time. Therefore, continued attention with regard to matching these motivations is crucial.

Insights in scientist motivation are inconsistent and moreover, they cannot be translated one-to-one to practitioners or (local) government representatives for two reasons. First, scientists are concerned with scientific data collection (Rotman et al. 2012), while practitioners are often interested in improving management practices (Weng 2015) and government agencies are concerned with policy making (Hollow et al. 2015). Second, the different role of authorities leads to different

expectations. Water authorities believe that citizens see water management as a task for authorities only, which implies that citizen do not want to be involved. Nevertheless, most water authorities agree that they need the observations of citizens for their work (Wehn & Evers 2014). Government practitioners have shown an interest in citizen science, but applying citizen science here is concerned with more risks compared to scientists. Government practitioners have a public role with different expectations. Increased insight in the motivations of government authorities can be used to enhance the design of citizen

science projects, tools and collaboration (Rotman et al. 2012).

This study aims to identify different viewpoints regarding citizen science among the water authorities in the Netherlands. Previous studies on this topic have often been limited to case studies, but this study aims to capture a general image of citizen science in water management in the Netherlands. The study seeks to determine which of the goals of citizen science

are supported by viewpoint and to determine to what extent water authority practitioners support higher levels of citizen involvement. Finally, the article will discuss the design implications for setting up citizen science projects at the water authorities.



## 2 Method

The research method used in this paper is based on Q methodology. It attempts to match the two methodological requirements set to identify individual perspectives at the Dutch region water authorities. The approach should be able to map the variety of viewpoints and to generalise this to a larger group. Q methodology was first introduced in 1935 by
Stephenson (Watts & Stenner 2012) and gained wider popularity after Brown's paper on political subjectivity (Brown 1980). The strengths of the Q methodology are that it combines qualitative and quantitative aspects and that it is statistically robust with small samples. Q methodology is applied to a small sample size of 30-40 people (Van Excel & De Graaf 2005; Watts & Stenner 2012), and the data is statistically analysed using factor analysis. Q methodology is a relatively uncommon method in science communication and water resource management (e.g. Raadgever et al. 2008), but it is a popular method used in
social sciences fields, such as political science and psychology (Cools et al. 2009).

Q methodology is used to describe a population of viewpoints using a factor analysis (Van Exel & De Graaf 2005; Watts & Stenner 2012). In the commonly used R methodology factor analysis, traits are variables and persons form the sample, while in Q methodology the variables are persons and the sample is formed by their opinions (Van Exel & De Graaf 2005). The
sample size is selected so that it includes a wide spectre of the existing discourses. Additionally, Q methodological approaches are abductive in nature by searching for explanations for empirical observations (Watts & Stenner 2012). Q methodology aims to describe and explain observations in order to develop theory (Watts & Stenner 2012).

Q methodological research is usually conducted in five stages (Van Exel & De Graaf 2005), which are summarised in Figure
1. Steps 1 to 4 are data collection steps, while step 5 consists of data manipulation and analysis. As a final endeavour each extracted factor array will be translated into a viewpoint.

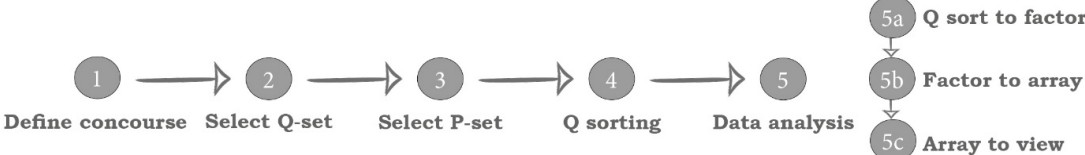

**Figure 1 – Flowchart of the five steps of the Q methodological approach (based on Van Exel & De Graaf 2005).**



### 2.1 Data collection

#### 2.1.1 Step 1: defining the concourse

First, we determined the concourse (i.e. all possible statements), which should contain all relevant aspects of the discourse (i.e. all discussions occurring on the topic) and represent all possible opinions. The concourse was formed by quotes taken

from the transcripts of: (a) ten semi-structured interviews with water authority employees, nature managers and citizen organizations, (b) a structured group discussion about citizen science with water professionals[1] and (c) a focus group meeting with five women who are part of an informal walking club in nature areas. The benefits and downsides of citizen science were extracted from the literature and common sense was used to supplement the concourse. Our concourse consisted of 229 statements about the application of citizen science in water quality monitoring at regional water authorities in the

Netherlands.

#### 2.1.2 Step 2: selecting the Q-set

Second, we compressed the concourse in two steps to a Q-set of 46 statements. A preliminary Q-set of 65 statements was formed by combining similar concourse statements and rejecting statements out of scope. This preliminary Q-set was tested by 6 master students between the age of 22 and 25. Two female students had a major in water resource management. The

other two female and two male students followed a major unrelated to the study topic. 19 statements were omitted or merged with other statements as a result of this trial. Table 2 contains the final set of statements.

#### 2.1.3 Step 3: selecting the P-set

Third, the P-set (i.e. the participants) was sampled using both a structured approach and snowball sampling. Initially, three criteria were used to select eight different water authorities: flood risk, age (expressed in years since the last reform or

merger with another water authority) and location (within or outside the urban conglomerate Randstad). Next, people were asked to suggest other colleagues with a different viewpoint in order to increase the diversity of opinions. Participants #20, #24, #25, #30 and #31 out of 33 were recruited with this strategy. Two to six people with different positions were interviewed per water authority, which resulted in interviews of one politician, 20 policy advisors, ten ecologists and hydrologists and two field staff members.

#### 2.1.4 Step 4: the actual Q-sorting process

Fourth, the Q-sorts (the actual arranging from) took place in four sub-steps:

---

[1] Part of the workshop 'citizen science' at the symposium 'De fysieke Digitiale Delta' [the Physical Digital Delta], which was organised by the Dutch water authority Delfland in 2014.





I. **Introduction to the research and the research method.**

Three examples of citizen science were presented to all participants in order to ensure that they have a basic level of understanding of citizen science. These examples were:

- the Dutch garden bird count (www.tuintelling.nl);
- iSPEX, a single event where citizens measured particulate matter with a smartphone device called iSPEX (Snik et al. 2014, p. 7351);
- a project of water level monitoring by citizens in a Dutch water authority (UvW 2015, p. 15).

II. **Pre-sorting of the statement**

The 46 statements were printed on separate numbered cards. The participant placed the numbered statement cards either on the agree, disagree or neutral pile. The purpose of this sort was to become familiar with the statements and make an initial division of the statements.

III. **Final sorting of the statements**

Participants ranked the statements according to a fixed distribution with the shape of a normal distribution to allow for factor analysis, see Figure 2. For example, the pile with +4 (most agree) was allowed to contain two statements only.

IV. **Post-sort interviewing**

In a structured interview, participants were asked to explain their Q-sort. Each participant explained their reasoning behind placing the statements in categories +4 and -4 and (if time allowed) any statement of their choice.

Post-sorting interviews were included in this study, because they can provide in-depth insight in to the beliefs and values underlying the sorts and allow for an analysis based on the participants' rationale rather than on the available literature or the researcher's bias (Gallagher & Porock 2010). It would have been preferred if all statements could have been discussed after sorting, but this required too much time for both researchers and participants.

**Disagree**          **Agree**

| -4 (2) | -3 (3) | -2 (5) | -1 (8) | 0 (10) | +1 (8) | +2 (5) | +3 (3) | +4 (2) |
|---|---|---|---|---|---|---|---|---|

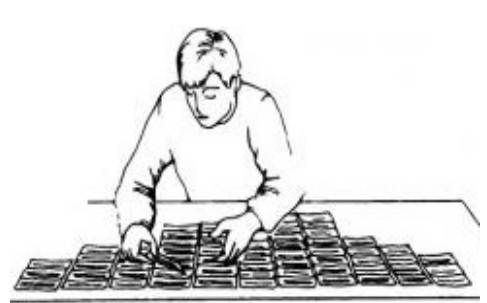

Figure 2 – The fixed distribution used in this study (left) and an impression of the sorting procedure (right).





## 2.2 Data operations

### 2.2.1 Step 5: analysis and interpretation

Fifth, the results were analysed and interpreted in three steps: (a) transforming the Q-sorts into factors following the approach described by Brown (1980) and Watts & Stenner (2012); (b) turning the factors into factor arrays; and (c) translating the factor arrays into factor interpretations (i.e. viewpoints) using the guidelines of Watts & Stenner (2012), Gallagher & Porock (2010) and Cools et al. (2009).

### 2.2.2 Step 5a: from Q-sort to factor

The factor analysis was performed using the commonly used software package PQMethod (version 3.2.1) (e.g. Van Exel & De Graaf 2005; Cools et al. 2009; Raadgever et al. 2008; Watts & Stenner 2012). Initial factors were extracted using the PQMethod's correlation matrix, relating the different Q-sorts to one another. The factor loadings of participants represent the extent to which their opinion matches the factor viewpoint. The final factors are obtained with an orthogonal factor rotation (Watts & Stenner 2012, p.118). The rotating process does not alter the results themselves, but changes the researcher's observation position in order to optimise the loading of each Q-sort on a single factor. A manual rotation was preferred above the built-in Varimax rotation of the PQMethod, because it has a lower inter-factor correlation which results in more distinct viewpoints.

The factors were assessed on their eigenvalue and the variance explained by the factor. High eigenvalues and high variance levels are associated with solid foundations for the study (Watts & Stenner 2012). An eigenvalue above 1.00 is generally considered sufficient, as it indicates the factor has enough in common with the other factors. It must be noted that the number of people loading on a factor gives an indication what portion of the participants share this viewpoint. This number cannot be used to determine the distribution of viewpoints in the total population without additional (quantitative) research.

### 2.2.3 Step 5b: from factor to array

Q-sorts with loadings that exceed the Significant Factor Loading (SFL) of 0.38 (based on Watts & Stenner 2012) were used to obtain an average ranking for each statement per factor. This weighed average results in the factor array, which reflects the ranking of an illusory person with a factor loading of 1.0 on this factor (see Table 1).

## 2.3 Data interpretation

### 2.3.1 Step 5c: from array to interpretation

The final factor arrays were interpreted using distinguishing items per factor and the post-sort interviews. We created a narrative of the +4 and -4 ranked statements and the statements ranked highest (lowest) by a factor, meaning this statement is





ranked lower (higher) in all other factors. This interpretation is thereby subjective in nature, although two mechanisms were applied to reduce researcher bias. First, the post-sorting interviews were conducted in order to be able to identify the underlying values and assumptions that enhanced the factor interpretations. Participant's afterthoughts were recorded, transcribed and categorised per statement and per factor. Second, all participants were presented an initial version of the narratives and were asked whether they recognised themselves in their assigned narrative viewpoint and if yes, why did they recognize themselves in this narrative.

## 3. Results

Based on the factor analysis, the 33 Q-sorts resulted in the identification of three factors that correspond to three viewpoints. The analysis of the factors justifies the inclusion of viewpoint C as a third, separate viewpoint. Factors A and B were convincingly included as distinguishing viewpoints. Factor C had an eigenvalue below the threshold of 1.00, but was included in the factor rotation. The distinguishing item analysis confirmed factor C as a third factor, since it has as many distinguishing items as factor A or factor B (see Table 1).

**Table 1 – Distinguishing items with the highest (left) and lowest (right) score per factor. Overlap indicates two factors had a similar ranking for an item, while the third factor ranked it different.**

|   | Highest | | | | Lowest | | |
|---|---|---|---|---|---|---|---|
|   | **A** | **B** | **C** |   | **A** | **B** | **C** |
| **A** | 7 | - | - | **A** | 9 | - | - |
| **B** | 5 | 10 | - | **B** | 5 | 11 | - |
| **C** | 6 | 3 | 10 | **C** | 7 | 4 | 7 |

The total variance explained by the Q-sorts should be above 35% (Watts & Stenner 2012), which is satisfactory with 53% variance explained. The factor arrays presented in Table 2 show how an individual would rank the items if that person were representing that viewpoint 100%. For example, statement 9 ("Citizen Science enables the collection of large amounts of measurements") would be placed in the most agree (column +4) by a person with viewpoint A, under agree (column +2) for viewpoint B and in the neutral (column 0) for viewpoint C.

The remainder of this section contains the three viewpoint narratives based on these factor arrays. Each of the three factors is presented as a viewpoint by creating a narrative out of the characteristic items and the most agree and most disagree statements. Item rankings are presented in the following format: (*item number : item ranking*) such that (2: +4) means item #02 is ranked +4 in this viewpoint. Interview fragments are integrated in the narratives as a quote followed by the letter Q



and a number indicating the source. For example ("quote" – Q1) means the quote comes from the Q-sort and thus participant

1. Figure 3 shows the availability of interview fragments per factor and per statement.

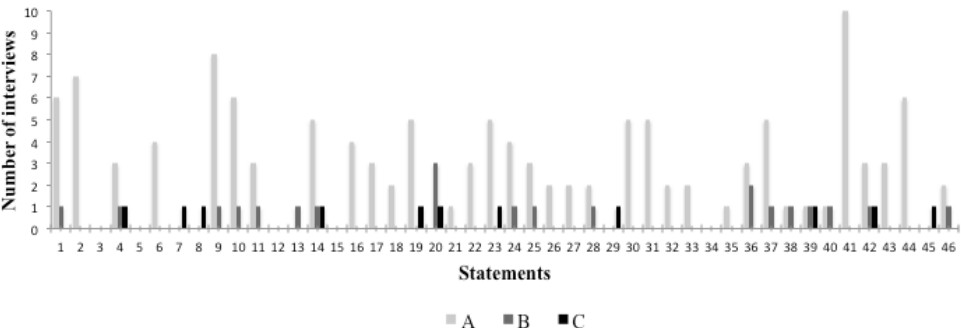

**Figure 3 – Distribution of interview fragments available per viewpoints and per statement.**

**Table 2 – Final factor arrays, the numbers in columns A, B and C are the theoretical item score for a person whose viewpoint is 100% that factor.**

|  | Item | A | B | C |
|---|---|---|---|---|
| 1 | Providing citizens with insight in water quality will only lead to unnecessary panic and questions. | -3 | -4 | -4 |
| 2 | Citizen Science is important, since it contributes to increasing water awareness. | +4 | +4 | +4 |
| 3 | Citizen Science is a solution to explain why you take certain measures as a water authority. | +1 | -1 | -1 |
| 4 | Water quality is an abstract concept, citizens will not understand what they measure. | -1 | -2 | -3 |
| 5 | It is important to have proper communications to citizens about why values deviate from the norm and what the uncertainty in the measured value is. | +1 | 0 | +1 |
| 6 | I would not know why citizens would not be interested in monitoring water quality. | -1 | -2 | 0 |
| 7 | Citizen Science is an economical way to collect (extra) measurements. | +1 | +1 | -1 |
| 8 | Citizen Science enables the collection of more measurements by conducting them more frequently. | +3 | +3 | +1 |
| 9 | Citizen Science enables the collection of large amounts of measurements. | +4 | +2 | 0 |
| 10 | Measurements and observations by citizens are no valuable addition to the official monitoring network. | -4 | -2 | -1 |





**Continuation of Table 2.**

| | Item | A | B | C |
|---|---|---|---|---|
| 11 | The most important goal is that the measurement data provides value to the water authority because the organisation has invested its time and energy. | 0 | +1 | +1 |
| 12 | I would rather make (smart) use of existing measurements than let citizens' conduct more measurements. | -1 | 0 | 0 |
| 13 | The greatest challenge is how to teach people something, if they can or want to spend little time on it. | 0 | 0 | -1 |
| 14 | Schools are especially suitable target groups to conduct these measurements, for example during a 'water lesson'. | 0 | 0 | 2 |
| 15 | The most important goal of citizen science is to teach people something about the environment they live in. | +1 | +2 | +3 |
| 16 | Citizen Science is an interesting social innovation, but not suitable for actually collecting useful data. | -2 | -2 | 0 |
| 17 | Citizens' abilities are often under estimated; they are better educated and smarter than we think. | +1 | +1 | 0 |
| 18 | As a water authority we need to learn how to handle the uncertainty of alternative (cheap) measurements that originate from Citizen Science. | +2 | +1 | +1 |
| 19 | Data collection by citizens is unreliable and should not be accepted by the water authority. | -3 | -2 | -1 |
| 20 | Citizens will only participate in Citizen Science, if participation is in their own interest. | 0 | +2 | -2 |
| 21 | Not all citizens can be trusted to conduct these measurements. | -1 | +1 | 0 |
| 22 | With a short training, citizens will be able to conduct measurements for the water authority. | 2 | +1 | 2 |
| 23 | Citizen Science is an interesting way to give meaning to the concept of citizen participation. | +3 | +1 | +3 |
| 24 | Citizen Science is necessary, because it helps to decrease the awareness gap between citizens and the water authority. | +2 | -3 | +2 |
| 25 | By using Citizen Science, the water authority shows that it is keeping pace with the times. | +1 | -1 | 0 |
| 26 | An important advantage of Citizen Science is that it reduces citizen's resistance to projects. | 0 | 0 | +1 |
| 27 | One can connect with and involve another part of the audience using Citizen Science. | +2 | 0 | +3 |
| 28 | As long as Citizen Science is not included in the policy at the top levels, the water authority should not invest in it. | -3 | -3 | -2 |





**Continuation of Table 2.**

| | Item | A | B | C |
|---|---|---|---|---|
| 29 | It is a major bottleneck to create support within the water authority for the deployment of Citizen Science. | -1 | 0 | -1 |
| 30 | The water authority will benefit from using Citizen Science in conducting its tasks, because less (financial) resources are available. | 0 | -1 | 0 |
| 31 | The conservative character of my organisation is a major bottleneck for Citizen Science. | -1 | -1 | -2 |
| 32 | The organisation is not equipped to work with large groups of citizen scientists. | 0 | +3 | 0 |
| 33 | My organisation has no capacity to work with all this data. | -2 | -1 | -2 |
| 34 | The water authority should incorporate in its policy how to deploy and stimulate Citizen Science more. | +2 | -1 | +1 |
| 35 | If citizens are structurally contributing, they should be compensated for that. | 0 | 0 | 0 |
| 36 | If citizens collect data for the water authority, they should have a say in the measures taken afterwards. | -2 | -4 | -3 |
| 37 | Citizens often have local knowledge and the water authority should use this knowledge. | +3 | +4 | +4 |
| 38 | Citizen Science is important, because it gives insight into the problems that citizens are concerned with. | +1 | 0 | +1 |
| 39 | Citizens should have insight in the most recent information of the water quality that is available with the water authority. | +1 | +1 | +2 |
| 40 | If you provide citizens with a reference framework, they themselves can validate their data. | 0 | -3 | -3 |
| 41 | I do not want citizens to interfere with our work. | -4 | -1 | -4 |
| 42 | The water authority should maintain control of conducting measurements, since the water authority is indeed responsible. | -2 | +3 | +2 |
| 43 | I think the creation of Citizen Science does not fall within the tasks of the water authority. | -2 | -1 | -2 |
| 44 | I do not have a full image of what is possible with Citizen Science. | -1 | 0 | -1 |
| 45 | An important caveat is that citizens will expect that their measurements will have a direct influence on policy. | 0 | +2 | +1 |
| 46 | Citizens cannot be motivated to participate in such projects for a long period. | -1 | +2 | -1 |



### 3.1 "Citizen participation for data application"

Factor A has an eigenvalue of 12.55 and explains 38% of the total study variance. 25 participants load significantly (i.e. loadings of 0.38 and above) on this factor. Among them are two people who load significantly on factor B as well and three that also load on factor C.

The people loading on factor A are a mixture of hydrologists, advisors, policy advisors, field staff and a politician. In this group are fourteen men and eleven women. Fifteen people are middle aged. They are distributed over all eight incorporated water authorities, which are located within and outside the Randstad and with a mixture of higher and lower flood risk. Six people work at a water authority that has been recently reorganized. In the following paragraphs, the factor is interpreted and

therefore referred to as a viewpoint.

### 3.1.1 Viewpoint A: "Citizen participation for data application"

Citizen science is important for water authorities to increase water awareness (2: +4), because citizens are unacquainted with the work of the water authority (*"People often do not know what the water authority is doing exactly and we do not really stand out. Citizens sometimes really wonder what they pay tax for [...]"* – Q5.) and *"what they can do themselves to improve*

*water quality."* (Q27)

Additionally, people with this viewpoint value citizen science for the collection of large amounts of measurements (9: +4) and for conducting measurements more frequently (8: +3). *"This data, it is an opportunity to have an area covering insight in dynamics of water quality and ecology."* – Q26. The organisation is expected to have sufficient capacity to analyse all the

data (33: -2) at the moment, but the water authority has to learn how to handle the uncertainty of these alternative (often more economical) measurements (18: +2).

These practitioners believe that their water authority should actively incorporate citizen science in its policy (34: +2) and that the water authority should not wait to invest in citizen science until it is included in top-level policies (28: -3).


They do not believe that the water authority needs to maintain control of monitoring, even though water authorities are in the end responsible for monitoring (42: -2). *"This is nonsense, because a lot is already measured by other parties."* – Q25. Employees do not fear citizen interference with their work (41: -4). They further believe that citizens, if provided with a reference framework, can validate their own data (40: 0). *"If they know what to do with it [the results], they can translate it*

*to their environment."* – Q11. Citizens can be trusted to conduct these measurements (21: -1). Although citizen science is a social innovation and the acquired data is less accurate, it should be accepted by the water authority (19: -3) (*"I mainly disagree strongly with the latter part of this statement."* – Q13 *[(...) and should not be accepted by the water authority]*) and



will be a valuable addition to the official monitoring network (10: -4). These people do not prefer the smart use of existing data to citizen science data (12: -1).

People with viewpoint A consider citizen science to be a solution when it comes to explaining why you undertake certain
measures (3: +1). This group further feels that citizen science will show that the water authority is keeping pace with the times (25: +1), although it is not a priority. Citizen science is an interesting way to give meaning to the concept of citizen participation (23: +3) and decrease the gap between the water authority and citizens (24: +2). They are not afraid that citizens will expect their contribution to have a direct impact on policy (45: 0) and they do not think citizens should get this influence (36: -2). *"You should prevent that, because manipulation [of results] is evident."* – Q9. These people believe that
giving citizens' insight in water quality will not lead to unnecessary questions and panic (1: -3). *"Those questions will come, but you should not be afraid, not afraid to say that you do not know everything."* – Q21.

### 3.2 "The water authority in control"

Factor B has an eigenvalue of 2.29 and explains 7% of the total study variance. Six participants load significantly on this factor (i.e. loadings of 0.38 and above). Among them are two people who load significantly on factor A as well and one that
also loads on factor C.

The people loading significantly on factor B form a mixture of advisors, policy advisors and field staff. Five out of six are male and four of them are middle aged. They work at three different water authorities, three people work outside the Randstad. Four people work at water authorities that have been recently (after 2005) reorganized. Four work in an area with
a high flood risk. In the following paragraphs, the factor is interpreted and therefore referred to as a viewpoint.

### 3.2.1 Viewpoint B: "The water authority in control"

People with viewpoint B consider citizen science to be important for increasing water awareness (2: +4). They believe that local knowledge will be valuable for the water authority (37: +4), as the citizen *"knows his own environment better than we do, on the small scale. We only have the broad overview in a large area"* – Q20. On the other hand, people with this
viewpoint strongly believe that the water authority needs to maintain control of monitoring, because they have the final responsibility (42: +3). *"I know what should be done with the data in the end. If we leave it to volunteers in this case, you have no reassurance on what comes when."* – Q3. Citizen Science allows for the collection of more measurements (9: +2) by conducting them more frequently (8: +3). This group does not believe the water authority needs citizen science to help fulfill its tasks to compensate for less financial resources (30: -1). It can also be noted that these people feel they do not have a full
idea (yet) of what is possible with citizen science (44: 0).





The water authority should not incorporate citizen science as part of its policy (34: -1), but on the other hand: they do think the water authority should invest in citizen science, even if it is not yet included in top level policies (28: -3). This group believes that a major bottleneck will be to create a support base within the organisation (29: 0).

They do fear that citizens cannot be motivated for a long-term participation (46: +2) and will not participate unless participation is in their own interest (20: +2) They further doubt whether all citizens can be trusted in doing these measurements (21: +1). *"If the citizen does not have personal interest, you have to wait to see what happens. Then he will think, I do not feel like it, I do not have time." – Q20.* This reflects their belief that most citizens would not be interested in participating (6: -2). People in this group do not fear questions or panic from citizens (1: -4). *"If you are so suspicious*

*towards your citizens, you have to question your role as government." – Q3.*

They think citizens will be able to conduct measurements after they receive a short training session (22: +1), but they do not expect that citizens will be able to validate their own data if provided with a reference framework (40: -3). *"If [data] quality is important to you, I am not sure whether citizens can do this." – Q18.* If citizens start collecting data for the water

authority, this group strongly feels that they should not be given more influence over measures (36: -4), but they do fear that citizens will think that their work will influence policy directly (45: +2). *"Citizens, I would almost say per definition, cannot do that [balance interests], they just want to do what they want." – Q30.* Also, they are convinced that their organisation is not equipped to work with large groups of citizen scientists (yet) (32: +3).

People with this viewpoint are not convinced that citizen science will involve another part of the public (27: 0) or that it is an interesting way to give meaning to citizen participation (23: 1). Moreover, they believe that citizen science should not be used to decrease the gap between citizens and water authorities (24: -3) or to show that the water authority is keeping pace with the times (25: -1). *"If this is your reason, I think it is rather cheap." – Q18.*

### 3.3. "Education and sharing local knowledge"

Factor C has an eigenvalue of 2.58 and explains 8% of the total study variance. Six participants load significantly on this factor (i.e. loadings of 0.38 and above). Among them are three people who also load significantly on factor A and one who also loads on factor B.

The people significantly loading on factor C are a mixture of advisors, policy advisors and field staff. Four people are middle

aged and three of them are male. They work at different water authorities, three within and three outside the Randstad. Three people work at a water authority that has been recently (after 2005) reorganized. Five out of six people with viewpoint C work in an area with a lower flood risk. In the following paragraphs, the factor is interpreted and therefore referred to as a viewpoint.





### 3.3.1 Viewpoint C: "Education and sharing local knowledge"

People in factor C think that citizen science is important, because it contributes to the increase of water awareness (2: +4). They also feel that the water authority should use the local knowledge that citizens have (37: +4).

Citizen should be given insight in the most recent information about water quality that is available with the water authority (39: +2). *"I believe that citizens and everyone have the right to get information from us." – Q10.* These people strongly disagree that providing citizens with insight in water quality will lead to unnecessary panic and questions (1: -4). They strongly reject the view that citizens should not interfere with their work (41: -4), although they believe the water authority should stay in control (42: +2). *"In my opinion information is essential for policy to be good, [so] I think they should be*
*collected by a professional." – Q10.*

The conservative character of water authorities is not considered a major bottleneck (31: -2). People with this viewpoint consider citizen science to be a good way to bind and involve another part of the audience (27: +3), to decrease the gap between citizens and water authorities (24: +2) and to a lesser extent, to reduce citizens' resistance to projects (26: +1). A
caveat could be that citizens will expect their measurements to have a direct influence on policy (45: +1), even though they should not be given a say in the measures taken afterwards (36: -3). *"For me these are two separated tracks. [...] they have this influence via the representatives that they can elect for the board." – Q31.*

They consider citizen science to be merely a social innovation, rather than a way to collect useful data (16: 0). This is also
reflected in their relatively small support of the idea that citizen science will allow for the collection of large amounts of data (9: 0) and the possibility of conducting measurements more frequently (8: +1). *"It is mainly supportive material and not a replacement of existing sources, because it is invalidated and uncertified information. I do not think that will fit" – Q31.* Moreover, people in this group believe that citizens will not be able to validate their own data (40: -3).

The most important goal will be to teach people something about their environment (15: +3) and especially schools will be a good target audience (14: +2). *"It is a good way to keep them [students] engaged." – Q10.* They further think that citizens will understand what they measure, even though water quality is an abstract concept (4: -3). *"I think this is an offensive comment toward the citizens, as if they are stupid." – Q10.* They believe it is possible to teach people something within a short period of time (13: -1) and they find it difficult to think of reasons why people would not be interested in water quality
(6: 0). They do think that citizens will participate, even if participation does not directly serve their own interests (20: -2). *"I participate as a citizen in a sort of science project, I do not do that for my own benefit, but because I like it and want to contribute. I think I am not the only one" – Q10.*





### 3.4 Evaluation of support for the three levels of engagement

The results indicate that there is a broad support base for contributory citizen science projects, but no support was found for collaborative or co-created projects, following the definitions of Bonney et al. (2009). All viewpoints support the collection of data by citizens, thus contributory projects. Viewpoint A is optimistic towards citizen participation in the analysis of the

data (see statement 40), suggesting the potential for collaborative projects. Viewpoint B and C are wary of involving citizens in these steps of the research process. None of the viewpoints supports statements related to co-created projects. Regarding the involvement of citizens in defining the question (see statement 38) participants said: *"There can be [topics] which we think they are important, while citizens do not find it important in the end."* (Q24, viewpoint A) and *"In that case you should answer all [these questions of citizens] and I think our organisation is not equipped at the moment"* (Q18, viewpoint B).

Regarding the translation of results to action (see statement 36) participants said: *"They [citizens] can only focus on the problems in their direct environment, but not on the implications for a wider area"* (Q19, viewpoint A); and *"I would not go that far"* (Q18, viewpoint B). External (e.g. the legal obligations regarding water quality monitoring of the water authority) and internal (e.g. procedures for citizens to influence decision making) conditions were mentioned as underlying causes in the post-sort interviews.

## 4 Conclusion

This paper sought to increase our knowledge about government practitioners' motivations and attitudes towards citizen science. A Q methodological approach was applied to identify the viewpoints of practitioners on citizen science, in the case of water quality monitoring at Dutch regional water authorities. 33 Water authority employees sorted a set of 46 statements related to citizen science. Three differing factors were identified in a factor analysis and translated into corresponding

viewpoint narratives.

The first viewpoint, viewpoint A, is named 'Citizen participation for data application'. People with viewpoint A see more opportunities than challenges when it comes to citizen science. They see applications in practical use of the data, but also for the active engagement of people. The second viewpoint, viewpoint B, is named 'Water authority in control'. People with this

viewpoint see a potential for data contributions by citizens in an illustrative way, but are concerned with challenges in organisational capacity, expectation management and motivating citizens as well. The third viewpoint, viewpoint C, is named 'Education and local knowledge'. People with this viewpoint focus on educational goals, such as teaching people about their environment and getting schools involved. They consider data applicability of secondary importance, although the data can be used illustratively. All three viewpoints are positive towards citizen science in the form of contributory

projects, where citizens collect data. Support for collaborative projects was found in viewpoint A, but none of the viewpoints support co-created projects between citizens and water authorities.





The outcomes of this study provide strong indications that practitioners at water authorities welcome citizen science in the form of citizen data collection. These practitioners believe citizen science can contribute to bridging the awareness gap as defined by the OECD (2014) and the Dutch Water Authorities (UvW 2015). Moreover, according to the participants, citizen science has the potential to transform governance structures, although the introduction of higher levels of citizen engagement will be challenging.

**5 Discussion**

In this study we mapped the opinions of practitioners at water authorities about using citizen science in water quality monitoring as a form of, among others, science communication and data collection. Three viewpoints were developed as a result of the Q methodological approach.

It must be noted that there is a rather high correlation between all three factors (see Table 3), which indicates that they are interrelated and overlap. The voluntariness of participation might have attracted participants with a positive attitude towards citizen science. The second sampling strategy recruited five participants that were expected to have different viewpoints. Two of them had viewpoint B and one had viewpoint C, thus enhancing the scope of viewpoints.

**Table 3 – Correlations between the final factor arrays A, B and C.**

|   | A | B | C |
|---|---|---|---|
| **A** | 1.00 | 0.26 | 0.43 |
| **B** | 0.26 | 1.00 | 0.35 |
| **C** | 0.43 | 0.35 | 1.00 |

The Q methodological approach has demonstrated itself as a useful method for identifying viewpoints in an explorative phase, but also for science communication research at a deeper level. Q methodology is an abductive research approach (Watts & Stenner 2012), which means that we tried to understand and explain the data rather than describe it or test a hypothesis. The viewpoint narratives and the interview results were based on the interpretation of the authors. This approach is subjective in nature, but the results were confirmed in responses to our request to validate the results. 13 out of 15 responses were a full identification with the assigned viewpoint and the other respondents placed minor remarks about an overlap between viewpoint A and C and challenges in practice. We can conclude that researcher bias was limited and did not significantly influence the results. The study is further expected to be representative of water authorities in the Netherlands, but it was limited to eight water authorities and the topic of water quality. It might be that there are other viewpoints that were not presented in this study. Repeating the study at a national scale, followed by an R methodological study to evaluate distributions, would justify the generalisation of the results to the Dutch water authorities as a whole.





### 5.1 Added value of post-sort interviews

The results demonstrate that post-sort interviews can reveal underlying values or assumptions that remained uncovered with the factor arrays only, as claimed by Gallagher & Porock (2010). Time limitations resulted in an unequal distribution of statements and viewpoints (see Figure 3). We are convinced that the full availability of interview fragments would have

resulted in even more exciting outcomes. A single fragment can represent individual remarks or explanations of a viewpoint, but a consistent image arises when multiple interviews are present. Four participants with viewpoint A literally mentioned the same reason to support statement 2, namely citizens' unfamiliarity with the tasks of the water authorities. The consistency in these cases with multiple fragments defends the assumption that even stand-alone fragments can be representative of the viewpoint. The five interview fragments for statement 36 revealed a difference in reasoning why

citizens should not have a say in the measures taking afterwards. Two participants with viewpoint A stressed that is would result in the manipulation of results, while three people with viewpoint B stressed the expertise of the water authority to balance conflicting interests and take an informed decision. Future research should consider allowing more time for post-sort interviews or organising group discussions to uncover the participant's underlying reasoning.

### 5.2 General impression of results on citizen science in water quality

The identification of these viewpoints has contributed to the scientific body of knowledge on citizen science. This study has relevance in practice as well, because it has captured the opinions of the Dutch water authorities about citizen science and because it has revealed some constraints in the design phase of citizen science projects.

A lack of trust in citizens, low intentions to use the citizen scientist's data and lack of support for higher levels of

participation might collide with citizens' motivations, especially for practitioners with viewpoint B or C. A relation of mutual trust is required as the basis for effective citizen science projects and prolonged contributions by citizens (Rotman et al. 2012). Viewpoint B reveals distrust in the commitment of citizens', citizens' intentions to participate and their capacity (see statements 20, 21, 22, 40 and 45). Another important motivation for citizens is the provision of feedback on how the data is used (e.g. Bonney et al. 2012; Rotman et al. 2012; Roy et al. 2012). Viewpoint C focuses on the goals of education

(see statements 14 and 15), with little emphasis on the actual use of the data (see statements 8 and 9). There is a lack of support for higher levels of participation in all three viewpoints. However, this is particularly the case for viewpoints B and C which do not support the idea of long-term projects where increasing levels of tasks and participation are required to keep citizens motivated (e.g. Rotman et al. 2012; Roy et al. 2012). The identification of these mismatches could be used to (re)design citizen science campaigns or to create a more balanced set of expectations that can guide citizen science projects.



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
