# Peer review of "A Q methodological approach to identify practitioners' viewpoints on citizen science in Dutch regional water resource management"

_Hydrology and Earth System Sciences, 2016_

## Referee Comment (RC1) · Anonymous Referee #1 · 13 Apr 2016

Review: A Q methodological approach to identify practitioners' viewpoints on citizen science in Dutch regional water resource management

Overview:

This paper considers an aspect of citizen science that seldom appears in previous studies, namely, how government practitioners view citizen science projects. The authors use a q methodology to summarize the views of the practitioners and reduce their opinions down to 3 main statements: Citizen science (1) encourages participation, (2) collects illustrative data, and (3) educates. The authors conclude that these results can guide future citizens science project planning.

[Figure]

General comments:

I am very interested in the topic this paper focuses on, and was particularly interested to hear how practitioners (connected with water resource management) in the Netherlands view citizen science projects. I was actually rather disturbed that the practitioners did not want to include citizen scientists in the co-design process. In my personal opinion, the co-design element is really important in citizen science. I believe that overall, the paper presents important results that I would very much like to see published in this special issue. Although I have strong reservations about the way the paper is written, these reservations do not reflect my feeling that the paper presents some important results. I hope the authors take my points as constructive criticism. I would very much like to see a revised version of this paper in the special issue.

Having said that these are important results, I do not agree with the authors that these results (in their present form) can be used to help future citizen science project planning. If they want to make this assertion, then they need to back it up considerably. This study has asked the people who would manage a citizen science project, what they feel about citizen science. Therefore, in my view, these results show how future citizen science projects WILL be planned. These results will not influence this planning. The managers will continue to do exactly what they have been doing up until now, which is exactly what the results reflect. If these results will instigate change or future planning, then something more has to happen. I would like to hear the authors' views on this.

My major concern is that of the audience, and the overall writing quality. I feel that this paper has been written with a statistically-experienced readership mind. I need to stress that this is a special issue for science communication. The issue will ultimately have a multidisciplinary audience from all fields within EGU and beyond.

As a disclaimer, if the authors had submitted this article to a statistics journal then I would not be qualified to review the methodology involved. However, as this is a

multidisciplinary special issue, I see my ignorance as a benefit, as I represent that target readership. I have no doubt that the authors have applied the q method properly and responsibly. However, the authors need to alter the article's focus and content so that is properly considers where it is going to be published.

Specific comments:

Firstly, the focus needs to change. I feel that the main focus of the paper should be on the results and not the method. This is after all a special issue about communication. I would suggest the focus should be shifted initially by simplifying the title to "The viewpoints of Dutch regional water resource management towards citizen science."

Continuing on the issue of focus, the explanation of the methodology is far too detailed at present and assumes a level of knowledge that I feel is more in line with an experienced statistics audience. Many terms that are not explained properly when they are first introduced. For example: factor analysis, extracted factor array, all the terms in Figure 1, PQMethod's correlation matrix, factor loadings, amount others. a more basic explanation of the method. It is also possible that the detail can remain if the authors are aware that explanations of terms are needed when they first refer to them.

It is not just the technical terms that are used without proper explanation. There are several instances in the Introduction alone that terms or ideas are introduced that then raise question marks. For example:

-"…citizen science as a potential solution for achieving various goals…" – What are these goals that the authors refer to? -"… no support for higher levels of citizen engagement…" – What are these higher levels? -"… identified viewpoints can be used to enhance the design of citizen science projects." – How? -"… citizens revel a "striking awareness gap" on water related issues." –What is this awareness gap? -"Citizen science is considered effective for a certain target audience…" –Effective as what? -"Insights in scientist motivation are inconsistent…" – Are you referring to scientists or citizen scientists? It's unclear who the main characters are throughout the

Introduction. -"The first is the short-term participation of volunteers. . ."- The first what? You are missing a noun here. Maybe use "Firstly,. . .." "Secondly,. . ." etc instead.

Secondly, I believe that the flow and general structure of the paper needs improvement. I accept that English is not the mother tongue of the authors. However, I need to stress that some basic rules of writing (in any language) need to be adhered to in this paper. My main concern is that of the paragraph. Each paragraph must (at least to some extent) focus around a single idea or topic. As an example, I would like to draw the authors attention to their 3rd paragraph in the Introduction. This paragraph introduces ideas of the growth of citizen science, "future challenges", a "striking awareness gap" (that needs explaining), and finally climate change and urbanization. It is quite possible that the authors want all this information in one paragraph, but if they do, then I highly recommend them to restructure to make the links clearer.

Continuing on the general structure, I think the sections can be arranged in a more intuitive manner. In particular, I'm afraid I very much dislike that the Conclusion comes before the Discussion. Both these sections start in almost exactly the same way, that it is hard to see the reasoning for this split and this order.

Finally, at the level of the sentence, the paper also needs considerable work. I have already mentioned that the paper needs to be re-tuned to the audience. This has knock-on effects with every aspect of the paper, especially at the sentence and word-choice level. Some examples of sentences that need changing: -"The study seeks to determine which of the goals of citizen science are supported by viewpoint. . ." -"Participants #20, #24, #25, #30, and #31 out of 33 were recruited using this strategy." – Do we need the actual participant numbers here? Why not just "5 out of 33"? -"High eigenvalues and high variance levels are associated with solid foundations for study" – How are eigenvalues and variances connected with solid foundations for study? And what is a solid foundation of study? -"The voluntariness of participation. . ." – I had to look up "voluntariness" to see if it was a real word. It is! However, it seems to be used mostly in legal settings. I have never heard it before and it made this reader

stumble hard. -"Factor A has an eigenvalue of 12.44 and explains 38% of the total study variance. 25 participants load significantly (i.e. loadings of 0.38 and above) on this factor." – This, and much of the results section, make no sense to me. These statements might make sense to q-methodology experts, but doubtful to the general scientific audience this special issue is aimed at.

The bottom line: Most importantly, I do not believe the authors need to re-do any of the analysis. I also think that the paper contains interesting and important results that should be published. However, the paper needs a complete overhaul in order to be appropriate for a science communication special issue. I have to be honest and say that, at present, the paper is not written well. The focus needs to change and become much clearer. There are countless times where terms/ideas are introduced without explanation. There are also many sentences that need improvement, and paragraphs that need structure. Every sentence, paragraph and section needs to be updated with the audience in mind. I would also suggest some professional type-setting or get some colleagues (from other disciplines) to review the paper thoroughly after the alterations.

I hope the authors take this challenge, as I would very much like to re-read this paper and fully understand what they have done. I look forward to reading the updated version.

---

## Referee Comment (RC2) · Anonymous Referee #2 · 26 Apr 2016

**1   General comments**

This manuscript analyses perspectives of practitioners related to the Dutch Water Authorities on the concept of citizen science. Citizen science as a topic is receiving increased scientific attention both from a natural and social sciences perspective. Where the former are mostly interested in the opportunities for data collection and analysis, the latter analyse aspects of motivation, dynamics of interaction, influence on decision making, empowerment, credibility, among others. Science communication and education are two very promising aspects of the potential of citizen science, and it is therefore opportune to analyse them in the context of this special issue.

But I identify 2 main weaknesses in the current manuscript. The first has to do with the focus, which balances uncomfortably between a methodological and a process oriented paper. More fundamentally, the manuscript takes a quite narrow view of citizen science, and by doing so in my opinion fails to present an in-depth and informative analysis. I elaborate on both aspects below:

1. The way that the manuscript is written, it seems to try to combine methodological novelty with generating insights in citizen science perspectives of professional practitioners: on the one hand it wants to prove that the Q methodology is useful to address this kind of problems, and at the same time wants to present insights in the perspectives of practitioners on the use of citizen science. Such combination is often not ideal. A methods paper focuses on improving a method, ideally by comparing it the outcomes of an established method as a benchmark. On the other hand, an insights oriented paper uses an established method to generate new insights on a certain process. To me, this manuscript would seem to fall in the second category: although I am not thoroughly familiar with the Q methodology, it seems a well-documented method that, while perhaps not very common in hydrology, is appropriate for this kind of problem and has been applied without much if any scientific novelty. In this case, the focus on the methodology can be reduced to a short argumentation of why the method is relevant and how it is implemented. That would allow for a clearer focus on the outcomes of the analysis, which I think is currently quite superficial and can be contextualised better (see point 2).

2. The manuscript is rather light, both on the contextualization of citizen science in the introduction, and the subsequent analysis and discussion of the results. The study seems to treat citizen scientists mostly as "assistants" to help solving a question (or generate relevant information) determined by the practitioner's reality. This is indeed a common form of citizen science but is arguably more relevant to citizen - scientist collaboration where there is a clear scientific objective (e.g., the seti@home project). In a policy context, as seems to be the case here, different forms of citizen science may

emerge that are not necessarily aligned to the agenda of the practitioner. Indeed, in the context of water quality the agenda of the citizen science may have a political nature that is not always aligned with government or may even be used to contest government practices or highlight deficiencies in governance (see e.g. Macknick and Enders, 2012, and the overview in Buytaert et al., 2014, table 1). It seems that the examples of citizen science projects used in this study are relatively "safe" from a political perspective (p4/4-6), but I can imagine that the perspectives of practitioners might be quite different if a more conflictive example of citizen science were to be included. The potential for such contestations to arise in this particular context is difficult to gauge without more background - indeed "conflictive" citizen science initiatives are more likely to emerge in regions with severe environmental issues and/or governance deficiencies, which is probably not the case here. But I hope that my point highlights the importance of the policy context, and related aspects/issues of buy-in, trustworthiness, and credibility in a citizen science - policy interface. Indicative of this need are also some conclusions drawn at the end but not elaborated much further. For instance, p6/1-2: "no support was found for collaborative or co-created projects": I wonder whether this is truly because of a lack of potential, or simply because of a lack of familiarity with collaborative citizen-science projects. Similarly, p17/3-4: "according to the participants, citizen science has the potential to transform governance structure". It is a pity that this is not elaborated further, because (as I have tried to argue above), aspects like these really touch upon some of the scientifically more challenging aspects of citizen science and its potential relation to policy.

Given the relevance of these aspects to the role of science communication, education, and more generally the science - policy - public interface, I would strongly encourage the authors to develop the manuscript along these lines, which I believe would increase significantly its relevance and impact.

[Figure]

**2 Specific comments**

1/8: "effective science communication": Citizen science is of course much more - see general comments.

1/19: "higher levels": might be explained further in the manuscript, but not very informative here. Can you be more specific?

2/17: an example of what?

2/21: "effective for a certain target audience and for certain levels of interaction": rather vague

2/22 - 25: quite anecdotal and short - can you elaborate more?

2/32: "Citizens engage in citizen science because they think it is fun": this is only one of many different reasons to participate in citizen science (see general comments)

3/27: aims to capture a general image of citizen science: Not sure what this refers to, especially because the reported study is also case study based.

3/30: "supported by viewpoint": Not clear. What viewpoint?

3/31: "design implications": this is not really elaborated much in the discussion/conclusions.

4/8: "relatively uncommon method": I suspect that the reason why it is uncommon is mainly because it is not applicable to typical problems in those scientific fields. Although I am not thoroughly familiar with the method, it would seem to a typical method for the type of question addressed in this study. If so, the description of the method can be short (relying instead on references); instead it is sufficient to explain how the method was implemented in this particular case (in order to ensure reproducibility). See also comments above.

5/13: "tested": how? More specifically, how were the 65 statements reduced based on

this test?

5/20: "Next, ...": How were the initial participants selected within the institutions?

5/20: "within or outside the urban conglomerate Randstad": I am not sure whether I understand this. Did you try to strike a balance between the number of institutions within and outside Randstad? If so, why?

5/22: "Two to six": why this diverging number?

5/26: "arranging from": what does this mean? Do you maybe mean "The actual ranking process"?

6/22: "preferred" -> preferable?

6/23: "too much time": so how was the interview 'reduced'?

8/5: "if yes": I am perhaps being a bit pedantic, but would it not be more informative to know if participants would not recognise themselves in the narrative?

8/10: What is considered to be "convincingly"? Is this an eigenvalue $> 1$?

8/14: "overlap": overlap between what?

9/5: table 3: Shouldn't the Y axis label read "number of interview fragments" instead of "number of interviews"?

12/11: subheader 3.1.1 is redundant because there is no 3.1.2. Remove it and perhaps change the title of 3.1 to that of 3.1.1. (the same for 3.2.1 and 3.3.1).

16/15: the conclusions should come after the discussion

17/12: "voluntariness" -> the voluntary nature

19/20: If this publication is forthcoming, surely the publication year cannot be 2014?

**3 references**

Macknick, J. E., and Enders, S. K. (2012). Transboundary forestry and water management in Nicaragua and Honduras: from conflicts to opportunities for cooperation. J. Sustain. Forest. 31, 376–395. doi: 10.1080/10549811.2011. 588473

Buytaert, W., Zulkafli, Z., Grainger, S., Acosta, L., Alemie, T.C., Bastiaensen, J., De Bièvre, B., Bhusal, J., Clark, J. Dewulf, A., Foggin, M., Hannah, D. M., Hergarten, C., Isaeva, A., Karpouzoglou, T., Pandeya, B., Paudel, D., Sharma, K., Steenhuis, T. S. Tilahun, S., Van Hecken, G., Zhumanova, M. (2014). Citizen science in hydrology and water resources: opportunities for knowledge generation, ecosystem service management, and sustainable development. Frontiers in Earth Science 2:26.

---

## Author Comment (AC1) · 13 May 2016

We would like to thank both reviewers for their extensive reviews. We noticed that the reviewers' comments overlap to a certain extent, thus we would like to address both reviews in one reaction. We are glad to hear both reviewers think our paper would suit the special issue, although requiring rewriting. Below we would like to respond to the comments made by both reviewers, first focussing on topics brought up in both reviews, before reacting to the individual reviews.

Reducing the 'general' method section, detailing the 'case specific' method description Both reviewers suggested that the method explanation is too extensive, albeit for different reasons. We opted for this extensive method description, because we assumed
not many hydrologists would be familiar with the method. However, it seems we over-done it. A method summary and a solid reference to an explanation of Q methodology will suffice. Doing so, we will reduce the amount of specific terminology of the method section, as was suggested by reviewer #1. We will only be able to reduce the statistical terminology to a certain extent, as we think it is important to provide the reader insight in the idea and the reasoning behind the method choices. Some terms and information (e.g. correlations between factors and eigenvalues of factors) are essential in doing so, as it will enable the reader to value the results in the discussion.

Reducing the method description will enable us to the case specific implementation of the method, as was suggested by reviewer #2. That way we create room to elaborate on the method, guided by remarks 5/13; 5/20; 5/22 and 6/23. In these remarks reviewer #2 rightfully asks for specifications on the method (process and choices made). Reducing the method explanation will further give us more space to discuss the insights, as suggested by reviewer 2.

*Design implications*

Both reviewers had a similar remark on our claim that this study enhances design of citizen science projects. These comments made us realise that we have not been clear about our definition of design. We extent our definition of design beyond the final design of a citizen science project, thus focussing on the process of designing the interaction between citizens and water authorities. This study was part of a larger, design-oriented research and in our enthusiasm we ill-defined what we mean by 'design'. We believe our study provides useful insights that can help water authorities to recognise view-points of their employees and incorporate these insights in the design process. We further speculated that they can subsequently match project managers with a certain viewpoint to project goals, target audience and project resources. But, as reviewer #1 pointed out: these results will not directly influence or change the final project design though, but merely help understanding why projects are designed as they are. In that respect they may be used to enhance the design process. In a revised manuscript we

should pay more attention to this.

*Conclusion/discussion*

Both reviewers suggested to have the discussion before the conclusion. In social sciences it is common to present conclusions, to discuss the meaning of these findings after. However, if it is in line with HESS we could discuss our results before coming to a conclusion.

*In response to reviewer #1*

Above we already responded to the issues of the design implications and the statistical and methodological-oriented nature of the paper. We also welcomed uggestions to match our text and content more to the target audience, such as simplifying the title and introducing terms before using them. The reviewer indicated to be unfamiliar with the terms of figure 1, but we hope the reviewer understands that using some statistical and method-specific terms is necessary. Although this special issue is aimed at non-experts in Q methodology, there may be readers familiar with the method. Therefore we would like the results to be complete (e.g. mention the eigenvalues and the level of significance). Using the suggested 'basic' or laymen description of these terms would only lengthen and complicate the explanation in our opinion. We do realise that we can improve figure 1 by adding a more self-explanatory caption.

We will introduce the concept of citizen science more thoroughly in the revised version. In the research we considered three levels of participation, nine goals related to citizen science and three types of governance, but we did not include them in the paper. In hindsight this was a pity, as incorporating these would have prevented the improper introduction of terms such as the 'goals' and 'higher levels of citizen engagement'. The awareness gap referred to is the lack of awareness of Dutch citizens on water issues in the Netherlands, their own influence on water (quality) management and the activities of water authorities. We will enhance the description of these and other concepts/ideas.
The paragraph level needs improvement and an example was given in the review (being 2/13-19; stating a growth in citizen science and causes for that growth). We struggle to split this particular paragraph as it only consists of 4 sentences, but believe that a proper introduction of concepts (see above) will solve the issue of paragraph structure.

The reviewer firmly states that he general structure needs thorough improvements, but limits the remarks to changing the order of the conclusion and the discussion. We think that a shift in focus from the method (Q methodology) to a focus on citizen science will improve the structure significantly. We propose to limit the method section to a brief introduction of key terms and focus on the relevance of the method for this case. Simultaneously, we will elaborate on the citizen science part, by properly introducing objectives that can be achieved with citizen science and by distinguishing clearly between different levels of citizen participation, based on existing citizen science literature. Both are used to structure the discussion, but more attention in will clarify the structure and readability of the article.

*In response to reviewer #2*

The reviewer indicated that the contextualisation and analysis of citizen science is rather shallow. We will not deny this. In the research we made a distinction between three levels of participation and three types of governance, but we did not include them in the paper. In hindsight this was a pity, as it would have clarified why we focus on a certain type of citizen science (with citizens as 'assistant').

The three levels of participation range from the 'assistant' (contributory citizen science) to citizens taking the initiative (co-creative citizen science). As the reviewer points out, citizen science may be 'conflictive'. Governance-wise we expect a water authority led governance for two reasons. As the reviewer already indicated, conflictive citizen science is more likely in areas with environmental or governance issues, which is not the case in the Netherlands. On the contrary: Dutch citizens lack awareness of water issues and are ignorant towards water management, reducing the likelihood of citizen

science initiated by citizens. The reviewer is right though, that this is not mentioned in our introduction and as a consequence, not addressed in the discussion. We can address this briefly in an updated version of the paper.

In this updated version we shall also specify how we implemented the method of Q. This update could include the following answers to remarks by reviewer #2: 5/13; The students performed the whole Q sort. During and afterwards they commented on individual statements. In a form they reported statements with an unclear formulation, statements similar to another and statements that were too broad for the topic. 5/20; We selected water authorities first. We approached either the communication department, people we had been in touch with before or the eco-hydrologists (because of the link with ecology, where citizen science is common practice). 5/20: Explorative interviews suggested a difference between urban citizens and rural civilians regarding attitude towards water authorities and water issues. Therefore we distinguished between the predominantly urban Randstad and the more rural non-Randstad. 5/22: The number of participants was aimed at four per water authority. Due to time constraints we were only able to visit some water authorities once. Some people cancelled our appointment (e.g. illness). To reach our total target of >30 participants we compensated at other authorities, hence the difference. 6/23: The interview was reduced by focussing on the +4 and -4 statements and 'statements of choice', as described in 6/16-18

Regarding the results: 8/5: We emailed participants, most replies only stated that they recognised themselves. Few elaborated, for example: 'viewpoint A is my ideal, but B is closer to reality'; 'I always look for collaboration and have a let's-go-for-it mentality, thus A is a match'; 'I think extra data is very welcome, thus I recognise myself in A'. 8/10: Convincingly is a combination of eigenvalues >1, having people with high factor loadings. People with viewpoint C relatively often had a significant loading on one of the other viewpoints as well. 8/14: 'overlap' refers to two statements being equally high/low ranked by two viewpoints (e.g. A & B), but clearly higher/lower than the third
(e.g. C). In the table these are all entries but the diagonals.

We would further like to thank reviewer #2 for pointing out inconsistencies in numbering, referencing and figures and for suggesting other literature.